# Free Vibration Characteristics of CFRP Laminate with One-Dimensional Periodic Structures

**DOI:** 10.3390/polym15051118

**Published:** 2023-02-23

**Authors:** Yukuan Dou, Jinguang Zhang, Xianglong Wen, Hui Cheng, Haixin Liu

**Affiliations:** 1School of Mechanical and Electronic Engineering, Wuhan University of Technology, Wuhan 430070, China; 2Sanya Science and Education Innovation Park, Wuhan University of Technology, Sanya 572000, China; 3Institute of Advanced Material and Manufacturing Technology, Wuhan University of Technology, Wuhan 430070, China; 4Hubei Provincial Engineering Technology Research Center for Magnetic Suspension, Wuhan 430070, China

**Keywords:** free vibration, carbon fiber reinforced polymer (CFRP), periodic structures, band gap

## Abstract

This paper proposes an approach of stacking prepreg periodically for carbon fiber-reinforced polymer composites (CFRP) laminate. This paper will discuss the natural frequency, modal damping, and vibration characteristics of CFRP laminate with one-dimensional periodic structures. The damping ratio of CFRP laminate is calculated using the semi-analytical method which combines modal strain energy with the finite element method. The finite element method is used to calculate the natural frequency and bending stiffness which are verified with experiments. The numerical results of the damping ratio, natural frequency, and bending stiffness are in good agreement with the experiment results. Finally, the bending vibration characteristics of CFRP laminate with one-dimensional periodic structures and traditional CFRP laminate are investigated with experiments. The finding confirmed that the CFRP laminate with one-dimensional periodic structures exists band gaps. This study provides theoretical support for the promotion and application of CFRP laminate in the field of vibration and noise.

## 1. Introduction

Vibration and noise universally exist in the mechanical system and affect the stability and reliability of the mechanical system, so it is important to reduce vibration and noise in various fields [1,2,3]. Carbon fiber-reinforced polymer composites (CFRP) have the advantages of high specific strength, high specific elastic modulus, and high damping in material properties [4,5,6]. There is a significant economic value for replacing metal materials with CFRP to achieve a lighter weight and to reduce vibration. The mechanics and vibration characteristics of structural parts made from CFRP can be designed through layer design, which has been widely applied in aerospace, transportation, engineering equipment, and other fields. For example, Yang et al. [7] applied CFRP to the raft frame of a ship and studied the relationship between the form of the raft frame structure and damping reduction, and proposed the damping reduction design principle of the carbon fiber composite raft frame. Zhang et al. [8] designed a CFRP motor shell, which reduced the vibration noise of the motor by 5 times more than the aluminum alloy frame motor.

Plates are one of the most widely used structural components. Researching plate components with excellent damping properties and reliable mechanical performance is important in reducing the vibration and noise of mechanical systems. Therefore, this paper provides an overview of the theory and application of carbon fiber composite damping reduction and structural reduction research to support the research in this paper.

In terms of the CFRP damping model; firstly, the complex stiffness model and strain energy model are most commonly used to establish the equation that connects with the damping loss factor [9]. Then, damping loss factors are solved through the finite element method [10,11], the analytic method [12], or the semi-analytical method [13].

For the damping properties of CFRP, scholars have studied the influence of ply angle, ply ratio and ply stacking sequence on the damping loss factor of CFRP laminate [14], and CFRP has been compounded with other materials to enhance the mechanical and damping properties of components [15,16,17]. Yang et al. [18] acquired the bending damping model of the cantilever beam based on the Euler Bernoulli beam theory, laminated plate theory, and energy method. Then, the accuracy of the analytical model is verified with the finite element method and by experiment. Scholars have studied the influence of the opening size and shape on the damping loss factor and natural frequency of CFRP laminate [4,19]. Kang et al. [20] looked into the effect of temperature on the modal damping coefficient of carbon-fiber-reinforced plastic materials and confirmed that the nonlinear characteristic of the modal damping was the lowest at a carbon fiber direction of 0 degrees. Gong et al. [21] discovered that the deposition of two-dimensional multilayer graphene oxide on the surface of the carbon fiber can enhance the damping properties of CFRPs in a wide range of temperature, frequency, and strain domains. The damping parameters of the graphene oxide interphase can predict the damping properties of multilayer carbon nanomaterial-modified composites under different service conditions. Qin et al. [22] found that carbon fibrous laminates with tailored carbon nanotube/polyurethane hybrid membranes possess good damping behaviors within a wide temperature range of 50~150 °C.

For the vibration characteristics of composite laminate, Lee et al. [23] studied the influence of the ply angle on the acoustic and natural vibration responses of CFRP laminate under the excitation of concentrated harmonic force. Kim et al. [24] proposed a free vibration theory that is derived using the Ritz method for rectangular CFRP laminates with holes, and investigated the effects of the design parameters on the vibration characteristics of the laminated composite rectangular plates with holes. Moumta et al. [25] who carried out a comparative study in the perspective of hygrothermal effects on free vibration characteristics of glass-fiber reinforced polymer (GFRP) composites and bamboo mat reinforced polymer (BMRP) composites. Xue et al. [26] established a vibration model using the Hamilton energy principle and looked into the vibration characteristics of moderately thick laminated composite plates with arbitrary boundary conditions. Chao et al. [27] performed calculations on the dynamic analysis of the jute fiber-reinforced composite single core and periodic core sandwich panels under different end conditions such as clamped-clamped, clamped-free, and simply supported-simply supported end conditions. Lv et al. [28] investigated the bending vibration characteristics of the piezoelectric composite double laminated vibrator under free and fixed boundary conditions by simulation and experiment. Zhang et al. [29] developed a unified model for vibration analysis of composite laminated sector, annular, and circular plate with various elastic boundary conditions, and the presented model exhibited fast convergence and good accuracy. A parameterization study on geometric parameters, material parameters, and boundary conditions was carried out to systematically reveal the vibration characteristics of rotary composite laminated plates.

The periodic structures have the band gap characteristic where the transmission of elastic waves can be prevented. Compared with active vibration isolation technology, periodic structures have the advantages of simple structure, low cost and are widely used to reduce vibration and noise. In a study of vibration characteristics of periodic structures made from CFRP, Iwata et al. [30] studied the wave propagation characteristics of the two-dimensional X-type periodic grid structure using the finite element method and the two-dimensional periodic structures theory, determining the relationship between the geometric parameters of the lattice structure and the filtering performance, and proposed a design method of the X-type periodic grid structure. Ren et al. [31] investigated band gap characteristics of a laminated composite beam with periodically placed piezoelectric actuator/sensor pairs and determined that the effects of the cross-ply angle of the laminated composite metamaterial beams and the structural parameters of the piezoelectric actuators and sensors on band gap. Li et al. [32] investigated dispersive behaviors and band gap characteristics of the multi-scale periodic composite plate that consists of an isotropic part and a laminated composite structure with grapheme platelets. Bishay et al. [33] proposed a design approach for laminated composite beams with periodic patches to obtain the patch length and stacking sequence that realizes the desired band gap frequencies.

At present, the research on CFRP laminate mainly focuses on secondary processing, surface damage, fatigue damage, impact energy absorption, mechanical properties, and damping and vibration characteristics [34,35,36,37,38,39,40,41,42]. Although the CFRP has excellent damping characteristics and the periodic structures have the bandgap characteristic, there is not so much research into the vibration characteristics of CFRP laminate with periodic structures. Therefore, combining the CFRP with the periodic structures to propose a CFRP laminate with periodic structures is important to improve the vibration reduction effect of engineering structural parts made from CFRP.

The remaining part of this paper is organized as follows. The CFRP laminate with periodic structures is introduced in Section 2. The modal strain energy method is used to calculate the damping loss factor of CFRP laminate in Section 3. In Section 4, the mechanical and free vibration characteristics of CFRP laminate with periodic structures and traditional CFRP laminate are compared. Some conclusions are given in Section 5.

## 2. Structure and Model

### Structure Statement

Figure 1 shows a schematic diagram of the CFRP laminate with periodic structures that contains three unit cells, and it is composed of the top face plate, bottom face plate, and central ply. The top and bottom face plates are made from continuous CFRP. The central ply 1 and central ply 2 with different ply degrees which are arranged three times along the length direction to form a CFRP laminate with three periods.

## 3. Damping Ratio

It can be figured out that in a vibration period the total energy loss of the structure can be considered as the sum of the strain energy losses in the fiber direction, the vertical direction, and the shear direction according to the Adams-Bacon damping model [43,44,45]. The specific damping capacity is the ratio of the dissipated energy to the total strain energy in a vibration period.
(1)$$\psi =\frac{\Delta U}{U}$$
where, *ψ* is the specific damping capacity, Δ*U* and *U* are respectively the dissipated energy and the total strain energy stored in a vibration period, *η* is the loss factor of the structure. The loss factor of the structure is converted into the damping ratio:(2)$$\zeta =\frac{\eta }{\sqrt{4+\eta^{2}}}$$
where, *ζ* is the damping ratio.

According to the constitutive relation of the composite laminate, the dissipated energy and the total strain energy in the cylindrical coordinate system are as follow:(3)$$\Delta U=\sum_{k=1}^{n}\sum_{i=1}^{3}\sum_{j=1}^{3}\eta_{ij}U_{ij}^{k}=\sum_{k=1}^{n}\sum_{i=1}^{3}\sum_{j=1}^{3}\frac{1}{2}\int \eta_{ij}\sigma_{ij}^{k}\varepsilon_{ij}^{k}dV^{k}(i,j=1,2,3)$$
(4)$$U=\sum_{k=1}^{n}\sum_{i=1}^{3}\sum_{j=1}^{3}U_{ij}^{k}=\sum_{k=1}^{n}\sum_{i=1}^{3}\sum_{j=1}^{3}\frac{1}{2}\int \sigma_{ij}^{k}\varepsilon_{ij}^{k}dV^{k}(i,j=1,2,3)$$
where, $\sigma_{ij}^{k}$ and $\varepsilon_{ij}^{k}$ (*i*, *j* = 1, 2, 3), respectively, represent the stress and strain components of the element *k*, among which 1 represents the fiber direction, 2 represents the direction vertical to the fiber direction, and 3 represents the direction vertical to 1–2 plane; *η_ij_* and $U_{ij}^{k}$ respectively represent the DLF and the strain energy component corresponding to the stress component $\sigma_{ij}^{k}$ of element *k*; *V^k^* represents the integral volume of element *k*.

According to the strain energy model(the Equations (1)–(4)), the strain components are extracted and input into the program written in MATLAB, as shown in Figure 2. The semi-analytical calculation process of CFRP laminates is as follows.

The equation for calculating the specific damping capacity is as follows:(5)$$\left[\begin{array}{ccc} u_{A11} & u_{A22} & u_{A12}\\ u_{B11} & u_{B22} & u_{B12}\\ u_{C11} & u_{C22} & u_{C12} \end{array}\right]\left[\begin{array}{c} \eta_{11}\\ \eta_{22}\\ \eta_{12} \end{array}\right]=\left[\begin{array}{c} \psi_{1}U_{1}\\ \psi_{2}U_{2}\\ \psi_{3}U_{3} \end{array}\right]$$
where *ψ*_1_, *ψ*_2_, and *ψ*_3_ represent the specific damping capacity of the first-order, second-order, and third-order modes, respectively. *U*_1_, *U*_2_, and *U*_3_ represent the total strain energy of the first-order, second-order, and third-order modes, respectively; *u_A_*_11_, *u_B_*_11_, and *u_C_*_11_ represent the strain energy of the first-order, second-order, and third-order modes along the fiber direction, respectively; *u_A_*_22_, *u_B_*_22_, and *u_C_*_22_ represent the strain energy of the direction vertical to the fiber direction, respectively; *u_A_*_12_, *u_B_*_12_, and *u_C_*_12_ represent the strain energy of shear direction in the 1–2 plane, respectively. Combine Equation (2) with Equation (1) to convert the specific damping capacity to the damping ratio.

### Numerical Modal Simulation of the CFRP Laminate with Periodic Structures

The unidirectional prepreg T700/SYE15001 consists of 64% T700 carbon fibers and 36% SYE15001 epoxy resin, and the thickness of one layer is 0.2 mm, the material properties are shown in Table 1. Modulus and density for unidirectional prepreg T700/ SYE15001 are provided by the manufacturer of the parent material. Damping loss factor reverse calculation [19] is used to determine damping loss factor of the unidirectional prepreg.

The Central ply stacking scheme of CFRP laminate with periodic structures are shown in Table 2. The configurations of specimens are shown in Table 3.

The software ABAQUS is used for the numerical modal analysis of the damping ratio. As shown in Figure 3, the finite element model of the CFRP laminate with a free-free end condition is built. The continuum shell SC8R elements are applied, and the sweep meshing method is adopted. The mesh size is 2 mm [19]. The frequency and damping ratio of the CFRP laminate obtained using finite element analysis (FEA) are shown in Table 4.

## 4. Experiment

### 4.1. Specimen Statement

As shown in Figure 4, Specimens were divided into two groups, and each group had three specimens. Z-1 was a CFRP laminate with periodic structures, and the period was 3 times. Z-2 was a traditional CFRP laminate with continuous fiber symmetrical laying. An autoclave process was used to fabricate specimens. The ply ratio (The ratio of the number of each ply angle to the total number of layers in each laminate) and structural parameters of laminates were the same, but the only difference between Z-1 and Z-2 was the ply stacking sequence. The influence of the periodic ply stacking sequence on the free vibration characteristics and mechanical properties of carbon fiber composite laminates was investigated.

### 4.2. Bending Stiffness of the CFRP Laminate with Periodic Structures

The three-point bending experiment was carried out on an electronic universal testing machine (DNS100 testing machine is procuded by CIMACH which is from Changchun, Chnia) to measure the stiffness of the CFRP laminate; the displacement error was less than ±0.5%, the measurement error of stiffness was less than ±1.0%. The applied load was 0–1000 N, the distance between support points was 240 mm, and the speed of the applied load was 5 mm/min. The experiment and simulation are shown in Figure 5.

The load and displacement curve after linear fitting is shown in Figure 6.

The equivalent stiffness of three points bending is calculated as follows:(6)$${\langle EI\rangle }_{eq}=\frac{Fl^{3}}{48\omega }$$
where *F* is load, N; *l* is distance between two support points, m; *ω* is flexure deformation, m.

Testing results of bending stiffness are shown in Table 5. The standard deviations of bending stiffness for group Z-1 and group Z-2 was 0.11 and 0.06, respectively. The bending stiffness results calculated by experiment and FEA were 5.06 × 10^7^ and 5.48 × 10^7^, respectively, which means an 8.3% difference for the CFRP laminate with periodic structures. For traditional CFRP laminate, the average bending stiffness was 6.40 × 10^7^, with a 7.8% difference between experiment and FEA. The simulation results were consistent with the experimental results. The bending stiffness of the CFRP laminate with periodic structures was reduced by 20.9%, compared with traditional CFRP laminate.

### 4.3. The Free Vibration Experiment

The differences in natural frequency, damping ratio, and frequency response characteristics between the CFRP laminate with periodic structures and traditional CFRP laminate in the free state were investigated. The experimental equipment mainly includes a computer, B&K3660C analyzer (It is produced by HBK, Moscow, Russia), accelerometer, and impact hammer, as shown in Figure 7.

#### 4.3.1. Modal and Damping Ratio Testing

In order to obtain the accurate results, the specimen length was divided into several parts as shown in Figure 8, and each cross point was taken as an exciting point. The exciting point was knocked three times, and the acceleration frequency response signals of all exciting points were fitted to get the free vibration frequency response curve. The natural frequency and damping ratio of the specimens were obtained using modal identification in BK connect 2018 software.

The modal testing system is shown in Figure 9. The input excitation signal was generated with the impact hammer. The output signals were detected by an accelerometer, and then transmitted to the B&K3660C analyzer. The vibration and noise analysis software in the computer was used to analyse the signals.

The testing results are shown in Table 6. The first-order, second-order and third-order natural frequencies were, respectively, 480 Hz, 1082 Hz, and 1260 Hz for the CFRP laminate with periodic structures, whose natural frequencies were less than Z-2. The mass, ply ratio, and structure size of Z-1 and Z-2 were the same, but Z-1 had lower stiffness than Z-2; therefore, the natural frequencies of Z-1 was lower. It can be found that the average damping ratios of Z-1 and Z-2 were 0.74% and 0.79%, respectively, whose average errors between FEA and experiment were 6.8% and 5.2%. Hence, the accuracy of the semi-analytical method for the CFRP laminate with periodic structures was validated. The Standard deviations for the natural frequency and damping ratio of specimens are shown in Table 7. The average standard deviations of the natural frequency and damping ratio were 7.14 and 0.017 for group Z-1. For group Z-2, the average standard deviations of the natural frequency and damping ratio were 6.16 and 0.018.

The mode shapes are shown in Table 8. The mode shapes of two kinds of CFRP laminates were the same. The first-order and third-order mode shapes were bending, the second-order mode shapes were torsional. The damping loss factor *η*_12_ was higher than *η*_11_ and *η*_22_, and the torsional mode shape produced more in-layer shear strain, so the damping ratio of the second-order mode was higher than the first-order and second-order modes.

#### 4.3.2. Band Gap Experiment

The bending band gap test is shown in Figure 10. The hammering method was adopted to measure the frequency responses in band gap experiments. The accelerometer was pasted at both ends of the specimen. A transverse impact force was applied on the CFRP laminate. Accelerometer 1 and 2 detected the input and output signals, respectively, and then transmitted them to the B&K3660C analyzer. The vibration and noise analysis software (B&K-connect) in the computer analyzed the signals. The window function of the B&K connect software was used to prevent signal leaks. The B&K3660C analyzer has a self-built anti-aliasing filter that provides a sampling frequency 2.56 times of the analysis bandwidth. The vibration test platform is shown in Figure 11.

The test results of bending vibration are shown in Figure 12. The band gap of periodic structure can be determined according to the input and output acceleration frequency response curve and the vibration level difference between the input and output acceleration.

It can be seen from Figure 12a, the output acceleration increased gradually with the decrease of input acceleration to 0–285 Hz. The bending vibration was magnified at 285–415 Hz. At 415–3000 Hz, the input and output acceleration have the same bending vibration characteristics, and there is no Bragg scattering phenomenon, Therefore, there is no bending band gap at 0–3000 Hz for traditionally laminated CFRP.

As shown in Figure 12b, the output acceleration decreased as the input acceleration increased, 435–592 Hz and 1120–1400 Hz, which was mainly due to the Bragg scattering phenomenon, that the input acceleration in these frequency bands was greatly shielded when it propagated through the periodic structures. So it can be judged that there are band gaps in these frequency bands. The maximum amplitude was attenuated by 5 dB and 10 dB in the first and second band gaps.

At 0–285 Hz and 0–140 Hz, the input acceleration decreased with the increase of frequency, but the output acceleration increased gradually; therefore, there is no indication that acceleration propagation was restrained. Only the viscoelastic damping of CFRP played a role in vibration attenuation, rather than the band gap.

## 5. Conclusions

In this paper, the CFRP laminate with periodic structures was presented. The damping ratio of CFRP laminate was calculated with the semi-analytical method. The modal and bending stiffness were simulated with ABAQUS, and the results were verified by experiment. The bending vibration characteristics of the CFRP laminate with periodic structures and traditional CFRP laminate were analyzed. The specific results are as follows:

(1) When ply ratio and structure size of laminates were the same, compared with traditional CFRP laminate, the natural frequency of CFRP laminate with periodic structures was lower, and their stiffness decreased by 20.9%. Average damping ratios of the CFRP laminate with periodic structures and traditional CFRP laminate were 0.74% and 0.77%; there was no obvious change in damping ratio.

(2) The CFRP laminate with stacking prepreg can periodically reduce vibration amplitude and block vibration propagation. In the free-free end condition, the CFRP laminate with periodic structures had bending band gaps at 435–592 Hz and 1120–1400 Hz, and the maximum vibration amplitudes decreased by 5 dB and 10 dB, but the traditional CFRP laminate did not have a band gap. The CFRP laminate with periodic structures in the free state had a damping effect on bending vibration at 0–140 Hz.

(3) We propose a kind of CFRP laminate with one-dimensional periodic structures, whose band gap characteristics can be applied in support beams and cantilever beams, etc. The influence of periodic layup scheme and different end conditions such as clamped–clamped, clamped–free, and simply supported–simply supported end condition on band gaps will be studied in the future. This paper provides a theoretical reference for the application of CFRP laminate with stacking prepreg periodically in the field of passive vibration reduction.

## Figures and Tables

**Figure 1 polymers-15-01118-f001:**
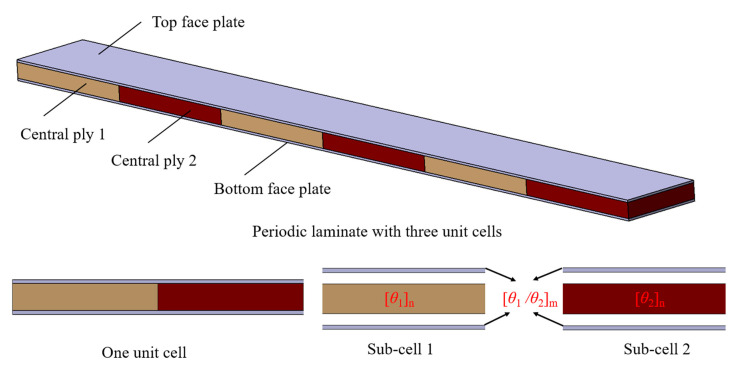
Schematic diagram of the CFRP laminate with one-dimensional periodic structures.

**Figure 2 polymers-15-01118-f002:**
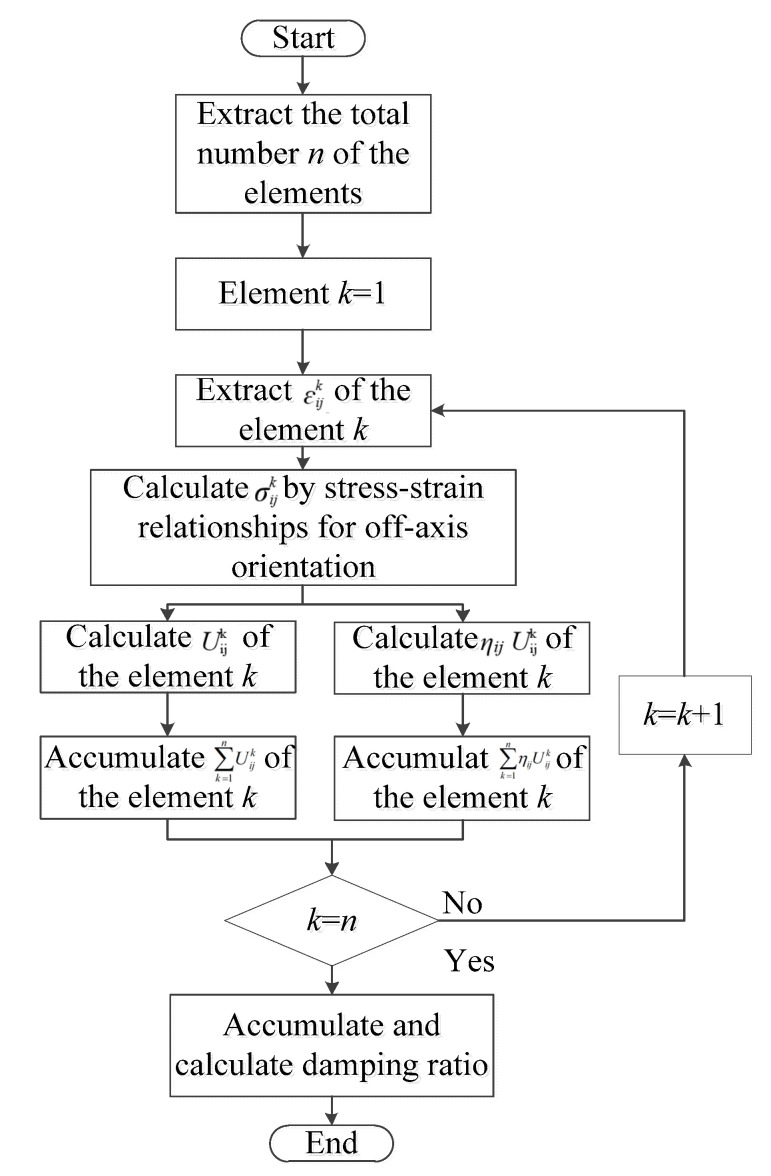
Process of the damping ratio calculation.

**Figure 3 polymers-15-01118-f003:**
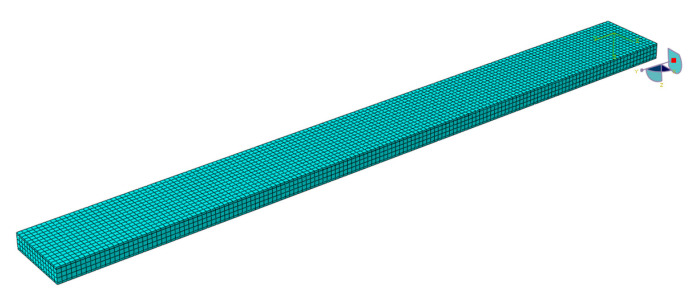
FEA modal of the CFRP laminate with periodic structures.

**Figure 4 polymers-15-01118-f004:**
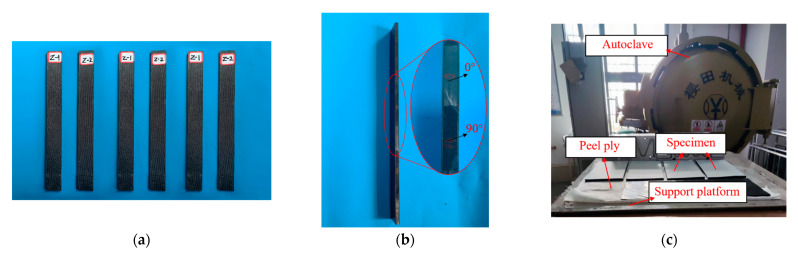
(**a**) Front view of specimens, (**b**) Side view of CFRP laminate with periodic structures, and (**c**) The specimen fabrication process.

**Figure 5 polymers-15-01118-f005:**
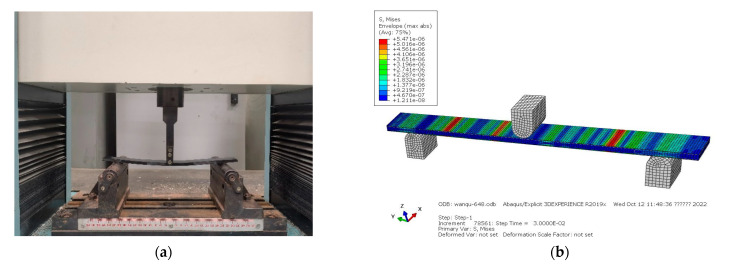
(**a**) Three-point bending experiment and (**b**) FEA model.

**Figure 6 polymers-15-01118-f006:**
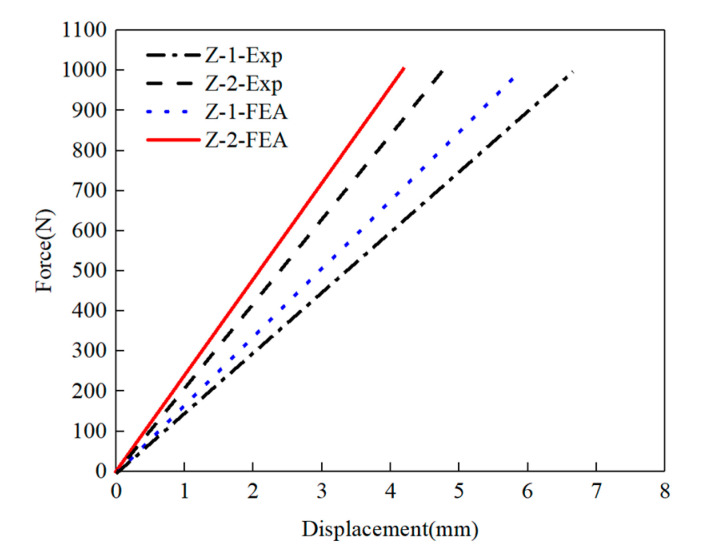
Load and displacement curve for bending stiffness.

**Figure 7 polymers-15-01118-f007:**
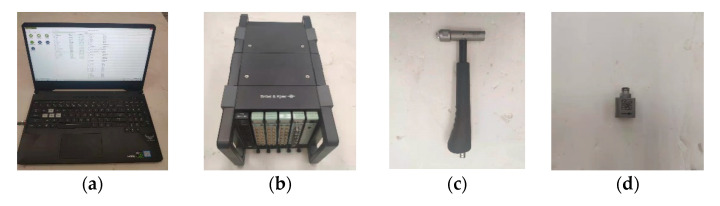
(**a**) Computer, (**b**) B&K3660C analyzer, (**c**) Impact hammer, and (**d**) Accelerometer.

**Figure 8 polymers-15-01118-f008:**
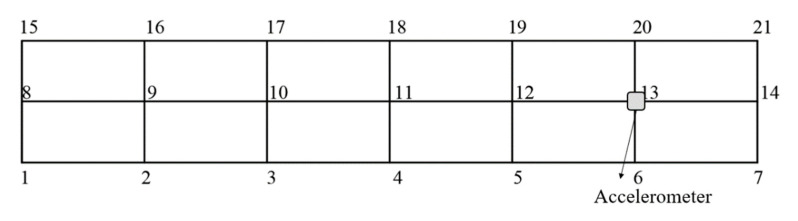
Exciting points division and accelerometer layout of the CFRP laminate.

**Figure 9 polymers-15-01118-f009:**
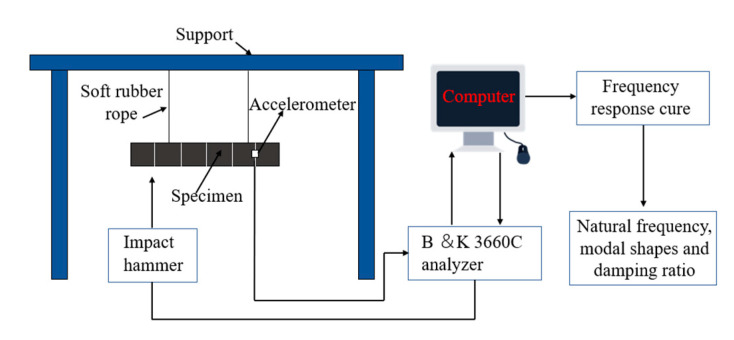
Modal testing system.

**Figure 10 polymers-15-01118-f010:**
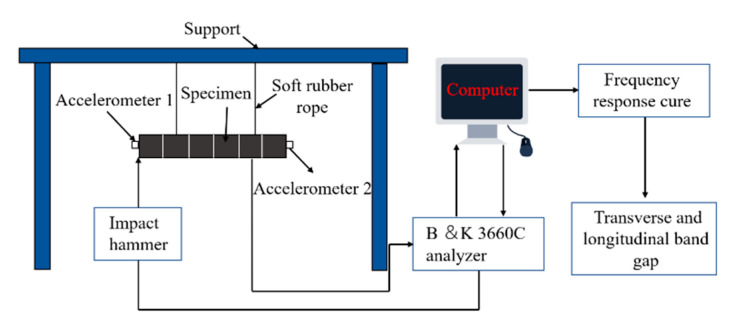
Band gap testing method.

**Figure 11 polymers-15-01118-f011:**
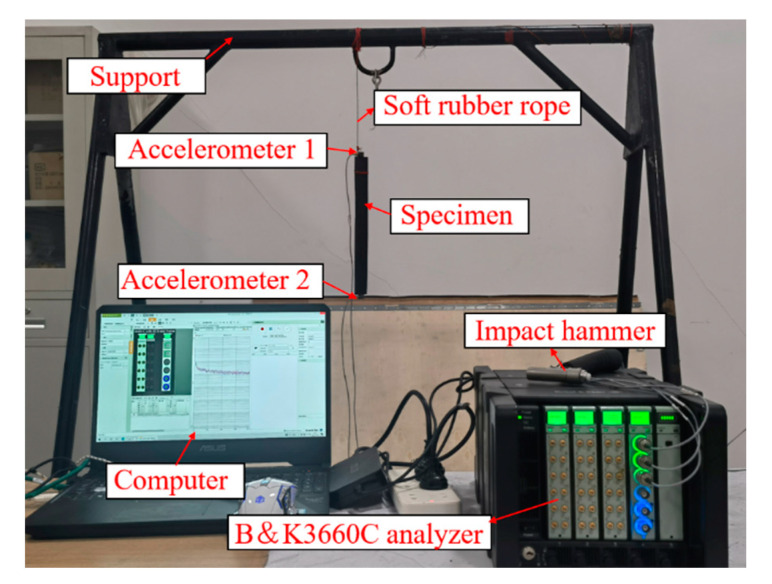
Free vibration testing platform.

**Figure 12 polymers-15-01118-f012:**
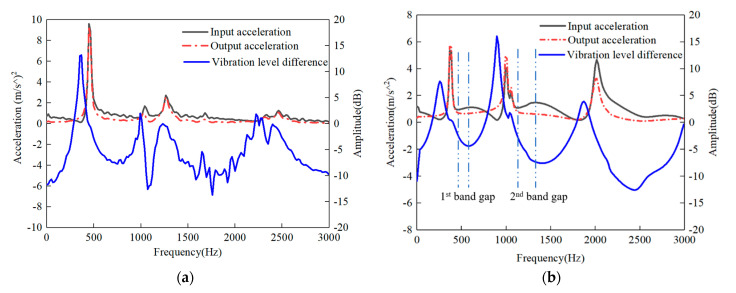
Frequency response curve obtained by experiment (**a**) traditional CFRP laminate (**b**) CFRP laminate with periodic structures.

**Table 1 polymers-15-01118-t001:** Material properties of CFRP T700/ SYE15001 prepreg.

Unidirectional Prepreg	*E*_1_ (GPa)	*E*_2_ (GPa)	*G*_12_ (GPa)	*υ* _12_	*ρ* (kg/m^3^)	*η* _11_	*η* _22_	*η* _12_
T700/SYE15001	142	9	4.6	0.32	1640	0.0031	0.0065	0.0125

Where *E*_1_: longitudinal modulus, *E*_2_: transverse modulus, *G*_12_: shear modulus in 1–2, *υ*_12_: Poisson’s ratio in 1–2, ρ: density, *η*_11_, *η*_22_ and *η*_12_: damping loss factor among which 1 represents the fiber direction and 2 represents the direction vertical to the fiber direction.

**Table 2 polymers-15-01118-t002:** Stacking method of the CFRP laminate with periodic structures.

No.	Length (mm)	Width (mm)	Thickness (mm)	Stacking
P-1	50	30	5.6	[0]_28_
P-2	50	30	5.6	[90]_28_
Top face plate	300	30	0.8	[0/90]_2_
Bottom face plate	300	30	0.8	[0/90]_2_
Central ply	300	30	5.6	(P-1/P-2)_3_

(P-1/P-2) represents a periodic unit arranged along the length of a rectangular plate. P-1/P-2 indicates that the periodic unit contains two components. The number of periods is represented by 3.

**Table 3 polymers-15-01118-t003:** Configurations of specimens.

No.	Length (mm)	Width (mm)	Thickness (mm)	Stacking
Z-1	300	30	7.2	[0/90/0/90/C_5.6_]_S_
Z-2	300	30	7.2	[0/90]_9S_

Where C_5.6_ indicates that the thickness of the central ply is 5.6 mm.

**Table 4 polymers-15-01118-t004:** Frequency and damping ratio of CFRP laminate obtained using FEA.

No.	Mode Ⅰ		Mode Ⅱ		Mode Ⅲ	
	Fre (Hz)	Damping Ratio (%)	Fre (Hz)	Damping Ratio (%)	Fre (Hz)	Damping Ratio(%)
Z-1	505	0.57	1176	1.22	1359	0.57
Z-2	555	0.48	1189	1.22	1497	0.48

**Table 5 polymers-15-01118-t005:** Testing results of bending stiffness.

Group	No.	Mass (g)	Bending Stiffness (N·mm^2^)		Average Stiffness (N·mm^2^)	Standard Deviation	Error
			FEA	Experiment			
Z-1	1	94.3		5.06 × 10^7^			
	2	94.1	5.48 × 10^7^	4.93 × 10^7^	5.06 × 10^7^	0.11	8.3%
	3	94.4		5.19 × 10^7^			
	4	95.2		6.48 × 10^7^			
Z-2	5	94.6	6.90 × 10^7^	6.32 × 10^7^	6.40 × 10^7^	0.06	7.8%
	6	95.1		6.40 × 10^7^			

**Table 6 polymers-15-01118-t006:** Testing results of the free vibration.

Group	Mode	Natural Frequency (Hz)		Error	Damping Ratio (%)		Average Damping Ratio (%)		Average Error
		FEA	Experiment		FEA	Experiment	FEA	Experiment	
Z-1	1	505	480	5.2%	0.57	0.44	0.79	0.74	6.8%
	2	1176	1082	8.7%	1.22	1.35			
	3	1359	1260	7.6%	0.57	0.42			
Z-2	1	555	559	0.7%	0.48	0.46	0.73	0.77	5.2%
	2	1189	1134	4.9%	1.22	1.42			
	3	1497	1452	3.1%	0.48	0.44			

**Table 7 polymers-15-01118-t007:** Standard deviation of testing results.

Group	No.	Natural Frequency (Hz)			Damping Ratio (%)		
		Mode			Mode		
		1	2	3	1	2	3
Z-1	1	483	1080	1252	0.42	1.33	0.42
	2	487	1091	1270	0.47	1.36	0.44
	3	470	1075	1258	0.44	1.37	0.41
	Average	480	1082	1260	0.44	1.35	0.42
	Standard deviation	7.26	6.68	7.48	0.021	0.017	0.014
	Average Standard deviation	7.14			0.017		
Z-2	4	562	1135	1447	0.49	1.42	0.43
	5	565	1140	1462	0.44	1.4	0.46
	6	550	1127	1449	0.45	1.45	0.44
	Average	559	1134	1452	0.46	1.42	0.44
	Standard deviation	6.48	5.35	6.64	0.022	0.021	0.012
	Average Standard deviation	6.16			0.018		

**Table 8 polymers-15-01118-t008:** Mode shapes of specimens.

Mode	Z-1		Z-2	
	FEA	Experiment	FEA	Experiment
1	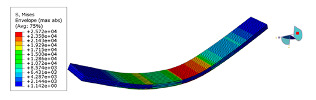	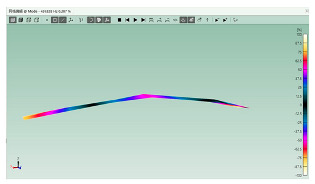	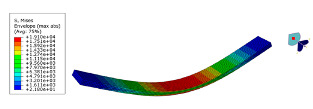	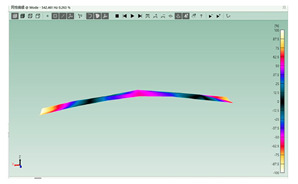
2	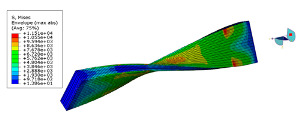	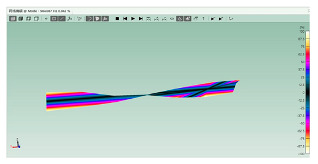	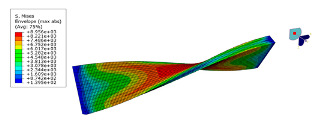	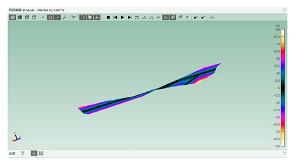
3	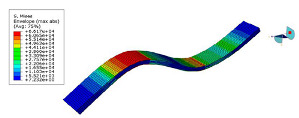	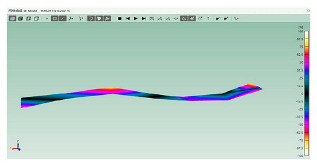	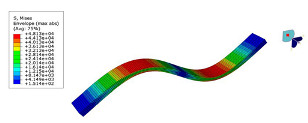	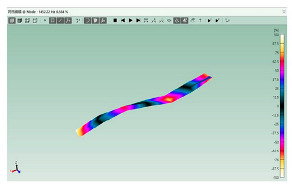

## Data Availability

The data presented in this study are available on request from the corresponding author.

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
