# Peer review of "Free Vibration Characteristics of CFRP Laminate with One-Dimensional Periodic Structures"

_polymers, 2023, doi:10.3390/polym15051118_

Round 1
Reviewer 1 Report
The manuscript entitled “Free vibration characteristics of CFRP laminate with one-dimensional periodic structures" addresses the stacking sequence, natural frequency, modal damping, and vibration characteristics of a CFRP laminate. Overall, the article requires addition of more literature discussions and free vibration result validation.
1. The specimen fabrication process with actual images to be added in the manuscript.
2. Introduction to be elaborated with more literature discussions on the vibration analysis of composite laminates.
3. The mode shapes presented in the table 7 are not legible enough to be published.
4. In experimental part, only the equipment’s used for testing are visible and the setup image provided in Figure 11 is not up to the journal publication quality.
5. The authors have not justified the influence of other ply orientations which can also be considered in the ABAQUS FE simulation through a parametric study and boundary parameters like clamped free and clamped-clamped conditions.
6. Literature validation is an important step in any scientific research, the manuscript lacks this criteria. It is recommended to validate the previous FE simulation results from the available literature data.
7. It is recommended to explain the particular engineering application of these CFRP laminate with one-dimensional periodic structures.
8. In the conclusion section, sufficient amount of result data are not critically discussed it resembles to be very short in the current manuscript.

Reviewer 2 Report
1. The introduction is too simple to describe the current development and challenges in this field.
There is no information about periodic structure and its advantages over traditional structures.
2. The introduction is not very comprehensive. Besides, the following recent periodic structure references are recommended to be cited: https://doi.org/10.1016/j.istruc.2021.08.071. https://doi.org/10.1080/15440478.2022.2025980
3. Where is the practical application of periodic structures? It must be added.
4. Draw the schematic picture for periodic structures for better visualization of readers.
5. For a better conclusion of the experimental results, the authors should have at least 3 specimens. Is it possible to introduce more tests to verify the repeatability of the manufacturing process?
6. Table 1 modify the damping loss factor percentages into values.
7. Section 2, The top and bottom face plates made of which material.
8. What is the overall length, breadth, thickness and ply orientation of Z-2 CFRP structure?
9. There is no in-depth explanation for damping ratio calculation, elaborate damping ratio calculation in detail.
10. There is a spelling mistakes in table 6 check it
11. On page 8; why did the authors not simulate the cantilever end conditions of the specimens? Why did the authors only show the free-free end condition of the simulated and experimental results of structure Z-1 and Z-2?
Reviewer 3 Report
Good research article. Need some more clarifications and info.

Round 2
Reviewer 2 Report
The necessary modifications have been made, and the paper can now be accepted.
Author Response
Thanks for your comments and suggestions